# *Corynebacterium striatum* Prosthetic Joint Infection Successfully Treated with Long-Term Dalbavancin

**DOI:** 10.3390/microorganisms11030550

**Published:** 2023-02-21

**Authors:** Bo Söderquist, Thomas Henningsson, Marc Stegger

**Affiliations:** 1School of Medical Sciences, Faculty of Medicine and Health, Örebro University, 701 82 Örebro, Sweden; 2Department of Orthopedics, Örebro University Hospital, 701 85 Örebro, Sweden; 3Department of Bacteria, Parasites and Fungi, Statens Serum Institut, 2300 Copenhagen, Denmark

**Keywords:** *Corynebacterium striatum*, prosthetic joint infection, genome sequencing

## Abstract

Arthroplasty surgery is a common procedure that significantly improves quality of life. The most feared complication is prosthetic joint infection (PJI), which occurs more often following revision surgery. Staphylococci are the most prevalent bacteria in PJIs, although many other pathogens have been reported. We describe a case of PJI in a 75-year-old farmer following revision surgery caused by *Corynebacterium striatum*, an unusual agent which normally occurs in the normal human skin microbiota with perceived low pathogenicity. Following a cemented right-sided total hip arthroplasty in 2006, a one-stage revision due to an osteolytic process in the right femur took place in 2020 with negative intraoperative tissue cultures. Three weeks later, the patient presented a fulminant infection which was treated with debridement, antibiotics, and implant retention (DAIR). Tissue biopsies showed *C. striatum* in 6/6 samples including small colony variants. Genome sequencing showed that all isolates differed by ≤6 SNPs with the same gene content related to resistance (*tet*(W) and *erm*(X)). The patient was sequentially treated with vancomycin, linezolid, and daptomycin, but due to side effects, treatment was changed to 12 weeks of dalbavancin as a 1000 mg loading dose followed by 500 mg intravenously/week. Impaired renal function during vancomycin treatment was normalized, and >1 year after finishing antibiotic treatment the outcome was still favourable. In conclusion, a case of a fulminant early post-interventional PJI due to *C. striatum* was successfully treated with DAIR and long-term dalbavancin therapy without any adverse reactions.

## 1. Introduction

Arthroplasty surgery is a common procedure that significantly improves quality of life for many patients. Osteoarthritis is the most common cause of worn-out joints that need replacement with an implant, and an aging population thus leads to an increased need for prosthetic joint surgery. The expanding number of patients with long-term arthroplasties in turn creates increased demand for revision surgery due to loosening of prosthetic devices/components.

The results of primary total hip and knee arthroplasties are usually excellent, with restored joint function and pain relief. However, serious complications such as deep infections occur in about 1–2% of cases [1,2]. Furthermore, the infection rate following revision surgery is significantly higher [3], ranging from 5% to 9% [4]. A prosthetic joint infection (PJI) often causes long-term suffering for the patient as it generally leads to repeated surgery, long-term antibiotic treatment, and significant disability as a final outcome. Besides the increased morbidity and mortality [4], there are significant costs for healthcare providers due to repeated surgery and prolonged and recurrent hospital stays [5].

The most common bacteria that cause PJIs are staphylococci. *Staphylococcus aureus* and coagulase-negative staphylococci (CoNS), predominantly *Staphylococcus epidermidis*, account for 55–75% of all PJIs when combined [6,7]. Infections caused by *S. epidermidis* are often associated with multidrug-resistant variants [8]. Other pathogens associated with PJIs include *Escherichia coli*, *Pseudomonas aeruginosa*, beta-haemolytic streptococci, enterococci, and anaerobes, especially *Cutibacterium acnes* [6].

*Corynebacterium* spp., particularly *C. striatum*, are reported as rare cases of PJI. However, the finding of this genus in tissue biopsy cultures is difficult to interpret, since it is generally regarded as non-pathogenic and a harmless commensal. In 2012, Cazanave et al. [9] described nine cases of PJIs where *Corynebacterium* spp. were isolated and determined to species level by molecular methods, but none of them were determined as *C. striatum*. Three cases of monomicrobial hip and knee PJI caused by *C. striatum* were later reported from the same group [7]. A report from 2018 by Kalt et al. [10] concerning the spectrum of *Corynebacterium* spp. isolated from orthopaedic infections included only four patients with PJIs, one of which (a chronic knee PJI) was considered to be caused by *C. striatum.* In 2019, Noussair et al. [11] published a retrospective study of monomicrobial bone and joint infections caused by *C. striatum*; this included nine episodes of PJI, a majority of which were successfully treated with the combination of amoxicillin and rifampicin. More recently, a database study by Tai et al. [12] on the microbiology of 2067 episodes of PJI revealed *Corynebacterium* spp. in 105 cases, including 32 early PJIs. In 31 of the 105 cases, the infection was a monomicrobial. However, the *Corynebacterium* spp. were not determined to species level in that study.

*Corynebacterium* spp., including *Corynebacterium striatum,* constitute an important part of the normal human microbiota of the skin and mucous membranes [13]. They are regularly isolated from superficial skin and soft tissue cultures, and are generally regarded as contamination since their pathogenic potential is considered to be limited.. However, they have been described as emerging pathogens, especially in patients with various indwelling devices and in immunocompromised patients [14], but also as noteworthy pathogens in bone and joint infections [15]. Since multidrug resistance is often present, treatment can be challenging. *C. striatum* often displays resistance to antibiotic groups such as beta-lactams, macrolides, lincosamides, fluoroquinolones, trimethoprim/sulfamethoxazole, and aminoglycosides [14,15]. Recommended antibiotics include glycopeptides, oxazolidinones, and daptomycin. However, lipidomic changes associated with daptomycin resistance in *C. striatum* have been reported [16].

We here present a case with a fulminant early post-interventional infection due to *C. striatum* following one-stage revision surgery that was successfully treated with debridement, antibiotics, and implant retention (DAIR), followed by long-term dalbavancin therapy.

### Case Presentation

A 75-year-old still-active male farmer with a history of atrial fibrillation and diabetes mellitus underwent a cemented right-sided total hip arthroplasty (*Lubinus*) due to coxarthrosis in 2006. In April 2018, the patient consulted his primary care physician due to arthralgia of the contralateral hip and suspicion of coxarthrosis. An X-ray was performed and en passant an osteolytic process, 2 × 9.5 cm, with a very thin corticalis of the right hip, was noted (Figure 1a).

The patient was referred to the Department of Orthopedic Surgery at Örebro University Hospital, Sweden, where further investigation by CT scan verified the thin corticalis and revealed a suspected breakthrough (Figure 1b). Ultrasound-guided biopsies of the osteolytic process were performed, and showed growth of *Staphylococcus hominis* in 1/5 samples, *Staphylococcus warneri* in 1/5, and *Corynebacterium* spp. in 1/5. The latter isolate was stored and subsequently in this study typed as *Corynebacterium minutissimum*.

Due to the obvious risk of a periprosthetic fracture, a one-stage revision arthroplasty surgery was performed in September 2020. Both the cup and the femoral stem were exchanged. An osteotomy was implemented in order to excise all the cement, but the thin corticalis was also fractured medially. Both the osteotomy and the fracture were repositioned and fixed by cerclage. Five perioperative tissue biopsies and synovial fluid did not show any growth. However, delayed wound healing and prolonged secretion were noted, and a CT scan 9 days postoperatively revealed encapsulated liquid not communicating with the joint, assumed to be a seroma or hematoma. The patient was discharged on day 11, but readmitted two days and again three days later due to wound leakage. This leakage was assessed as drainage of seroma or hematoma, since the patient was afebrile. It is possible that the leakage diminished temporarily, but on a scheduled early follow-up visit three weeks after surgery, profuse leakage of serous fluid and a phlegmonous process surrounding the surgical wound of the hip was noted (Figure 2).

A deep infection was suspected, and a DAIR procedure was performed the following day. Extensive necrotic tissue and pus were noted during the surgery. Thorough excision of the necrotic tissue, rinsing/lavage, and exchange of the femoral head were performed, and six tissue biopsies were taken for culture. Empirical treatment with intravenous cloxacillin and vancomycin was instituted. In the biopsies, growth of *C. striatum* was noted in 6/6 samples, *Staphylococcus hominis* in 1/6, and *Enterococcus faecalis* in 1/6, and so treatment with vancomycin was continued but cloxacillin was stopped. Due to profuse secretion and soft tissue inflammation, a renewed DAIR was performed 10 days later.

During the treatment with vancomycin, a steady increase of serum creatinine and decrease of glomerular filtration rate (GFR) was noted (92–139 μmol/L and 62–39 mL/min/1.73 m^2^, respectively). After four weeks, treatment was switched to oral linezolid. However, 14 days later the patient noted numbness and paresthesia in his hands, and treatment was again changed, this time to 700 mg intravenous daptomycin once daily (corresponding to 8.75 mg/kg). Four days later, the patient experienced myalgia and his serum creatine kinase increased from 0.8 to 20.4 μkat/L. Treatment with dalbavancin was then started, with 1000 mg as a loading dose followed by 500 mg intravenously each week. The patient was monitored, and a normalization of the serum creatinine (89 μmol/L) and GFR (63 mL/min/1.73 m^2^) was noted. The local status of the soft tissue improved very slowly, with persistent inflammation of the hip and thigh, and so the dalbavancin treatment was continued for 12 weeks.

At the end of treatment in February 2021, the patient’s CRP value was 4.0 mg/L and the inflammation of his extremity had significantly declined. The concentration of dalbavancin was determined after the first administered dose of 1000 mg, showing a trough value of 26.7 mg/L (Laboratoire de Pharmacologie Clinique, Centre Hospitalier Universitaire de Nantes, France). No additional determinations of the dalbavancin concentration were performed. At follow-up in December 2021, the patient’s CRP value had remained at <4 mg/L and his condition had improved. His functional status was good, and he used only one walking aid outdoors. As of September 2022, one and a half years after discontinuation of dalbavancin treatment, the patient was still infection-free.

## 2. Materials and Methods

### 2.1. Bacterial Cultures

Tissue biopsies (*n* = 5) and a synovial fluid sample obtained during debridement and revision surgery were per routine incubated (i) on GC agar (GC Medium Base, Becton Dickinson, Sparks, MD, USA, supplemented with 1% BBL IsoVitaleX Enrichment) and incubated in air with 5% carbon dioxide (CO_2_) at 36 °C, (ii) on fastidious anaerobic agar plates (4.6% LAB 90 Fastidious Anaerobe Agar, Lab M, Heywood, UK) supplemented with 5% horse blood (*v*/*v*) and incubated at 36 °C in an anaerobic atmosphere, and (iii) in fastidious anaerobic broth (2.97% Fastidious Anaerobe Broth, Lab M, supplemented with 1% D-glucose, VWR) incubated in air at 36 °C. The isolates were determined to species level by matrix-assisted laser desorption/ionization-time of flight MS (MALDI-TOF MS) (Microflex LT; Bruker Daltonik, Bremen, Germany) using Biotyper 3.1, and isolates were stored in preservation medium (trypticase soy broth with 0.3% yeast extract and 29% horse serum) at −80 °C.

Minimum inhibitory concentration (MIC) was determined by gradient test (Liofilchem, Roseto degli Abruzzi, Italy and Etest, bioMérieux, Marcy l’Etoile, France, respectively) according to EUCAST guidelines (https://www.eucast.org/ast_of_bacteria/; accessed on 1 March 2022). Antibiotic susceptibility testing was performed on Mueller–Hinton II agar 3.8% (*w/v*) plates (BD Diagnostic Systems, Sparks, MD, USA) at 36 °C.

### 2.2. Population Analysis Profile–Area under the Curve (PAP-AUC) Method

The method for the population analysis profile–area under the curve (PAP-AUC) was performed according to Wootton et al. [17] with minor modifications; the isolates were subcultured on Mueller–Hinton II agar plates at 36 °C. A 0.5 McFarland bacterial suspension was generated for each isolate, from which 10 μL was pipetted onto eight BHI agar plates containing different concentrations of dalbavancin (0, 0.064, 0.125, 0.25, 0.5, 1, 1.5, and 2 μg/mL). The droplet was manually spread across the plate. A swab from the 0.5 McFarland bacterial suspension was also incubated on each plate by spiral plating. The plates were then incubated at 36 °C for 48 h and numbers of colonies were counted manually. The isolates were tested in triplicate [18].

The results from the PAP-AUC method were processed as follows. The total number of colony-forming units (CFU) growing on each plate was counted and log_10_ transformed to obtain CFU/mL. The chosen maximum number of counted colonies was 1000, corresponding to 10^6^ CFU/mL. Each experiment was conducted in triplicate and plotted on a graph in order to calculate the AUC via the trapezoidal rule.

### 2.3. Genome Sequencing and Analyses

DNA was purified from four *C. striatum* isolates (2371 and 2374 from two different tissue samples, and two phenotypically different colonies from the sample 2371, respectively) using the Roche MagNA Pure 96 (F. Hoffman-La Roche Ltd., Basel, Switzerland) system after incubation overnight at 36 °C on blood agar plates (SSI Diagnostica, Denmark). Quantification was performed using a Qubit fluorometer (Invitrogen, Waltham, MA, USA), followed by library preparation using the Nextera XT DNA Library Prep Kit (Illumina Inc., San Diego, CA, USA) according to the manufacturer’s protocol with half volume. Sequencing was performed on a NextSeq 550 platform (Illumina Inc.) to obtain paired-end reads using a 300-cycle kit. The sequencing data were subjected to quality control using bifrost (https://github.com/ssi-dk/bifrost) to ensure adequate sequencing depth of both isolates prior to assembly using SPAdes v3.9.0. All genome sequences were archived at the European Nucleotide Archive under project ID PRJEB59823.

Phylogenetic analysis was performed by aligning the raw reads of the four isolates to the *C. striatum* strain 216 reference chromosome (GenBank accession number NZ_CP024931) using NASP v1.1.0 [19]. If a variant was present in <90% of the base calls per site per individual isolate or a minimum coverage of 10 was not met, the position was excluded across the collection to retain only high-quality variant callings. An additional SNP-based analysis was also performed, which included all publicly available (*n* = 26) *C. striatum* genomes from the NCBI RefSeq database (https://www.ncbi.nlm.nih.gov/refseq/, accessed on 1 August 2022) of human origin. ABRicate (https://github.com/tseemann/abricate) was used to search the assembled genomes for genes associated with resistance markers present in the ResFinder database [19].

## 3. Results

### 3.1. Clinical Course

The patient was treated with several potentially effective antibiotics, as described above, but because of adverse effects repeated exchange of antimicrobial agents was performed. Due to the long-lasting course and persisting signs of inflammation, a 12-week treatment regimen comprising 500 mg dalbavancin once weekly was finally administered. During the treatment, the status of the hip and thigh slowly improved, and laboratory parameters were normalized. With a follow-up of >1 year, the outcome of the PJI remains favourable so far.

### 3.2. MIC

The Corynebacterium striatum isolates displayed a MIC value of 0.032 mg/L for dalbavancin. The isolates were also susceptible to vancomycin (MIC: 0.5 mg/L), daptomycin (MIC: 0.064 mg/L), and linezolid (MIC: 0.125 mg/L), but resistant to gentamicin (MIC: 2 mg/L), benzyl-penicillin, ciprofloxacin, clindamycin, and tetracycline.

### 3.3. PAP-AUC

Two isolates (2371 and 2374) were subcultured on BHI agar plates containing different concentrations of dalbavancin as described above. The results of the PAP-AUC determination are shown in Figure 3. There was no indication of heterogenous intermediate resistance.

However, on BHI agar plates containing 0.125 mg/L of dalbavancin, growth of both normal-sized colonies and tiny colonies was noted (Figure 4). These colonies were subcultured, species determined by MALDI-TOF MS, and re-evaluated by MIC determination without any discrepancy. Isolates from the normal-sized and tiny colonies (IDs 2371 and 2374, respectively) were subjected to genome sequencing.

### 3.4. Genome Sequencing of C. striatum Isolates

Two *C. striatum* isolates (2371 and 2374) from two different tissue samples were stored and subsequently genome-sequenced using Illumina technology along with two phenotypically different isolates (both from sample 2371) identified during PAP/AUC testing; the sequencing thus included isolates from both normal-sized and tiny colonies (Figure 4). The number of SNPs differed by ≤6 SNPs across ~90% of the reference chromosome and with the same gene content related to resistance (tet(W) and erm(X)). Comparing the isolates to the available genome data of *C. striatum* highlighted a rather diverse population, with no observation of major lineages. While the four isolates included here were nearly identical to each other, they differed from all other isolates by some 15,000 SNPs, based on a core genome of 67% (2.0 Mb). A midpoint rooted maximum-likelihood phylogeny of all available *C. striatum* genomes of human origin in the NCBIs RefSeq database and the presented Swedish isolates are shown in Figure 5.

## 4. Discussion

The treatment of patients with early (or late but acute) PJIs is challenging, since eradication of the causative agent is required for success. This can be achieved by the DAIR procedure and efficient, biofilm-active antibiotics. Nevertheless, rates of successful outcome range from 54% to 83% [20], and are probably significantly lower following revision surgery. In the cases of failure or chronic PJIs, a one- or two-stage exchange surgery or lifelong suppressive antibiotic treatment would be required, and in the worst-case scenario a PJI may lead to resection arthroplasty, arthrodesis, or even amputation.

*C. striatum* is regarded as a commensal and of low virulence. However, the present patient presented a fulminant postoperative infection of a revised hip arthroplasty with cellulitis and drainage of the wound. DAIR was performed with the goal of avoiding a renewed and probably complicated surgical exchange. Multiple tissue cultures obtained during revision surgery showed growth of *C. striatum* in all samples. Due to multiple adverse reactions to the administered antibiotics (vancomycin, linezolid, and daptomycin), dalbavancin was finally chosen.

Dalbavancin has been approved for the treatment of bacterial skin and soft tissue infections as a two-dose regimen (1000 mg as a loading dose and one additional dose of 500 mg after 1 week) or as a single dose of 1500 mg intravenously. However, dalbavancin has also been considered for the treatment of conditions requiring prolonged antibiotic courses, such as joint and bone infections and cardiovascular infections with or without devices, as recently reviewed by Oliva et al. [21]. Considering PJIs specifically, there are still only a few reports [22,23], although in total hundreds of patients with PJIs have been included in other reports of bone and joint infections [22,24].

There are limited data regarding safety and efficacy for treatment beyond three doses of dalbavancin. However, in a study by Matt et al. [22], up to 10 doses of dalbavancin were administered for the PJIs besides suppressive treatment at 21-day intervals for 2 patients. In another study [25] of 101 patients, 32 of whom had PJIs, the median number of doses administered was 3 but the range was 1–32, and at least 1 patient with a PJI received 13 doses.

The optimal dosing and dosing interval of dalbavancin for extended treatment of bone and joint infections still remains to be defined [22]. Several preferential dosing regimens have been used [22,24]. Dunne et al. [26] found that in an extended-dosing study with 500 mg weekly administrations of dalbavancin, following loading doses, a steady state was reached without accumulation for 8 weeks. Data are accumulating that an initial dose of 1500 mg followed by a second dose of 1500 mg on days 7 to 14 should be administered, except for patients with severe renal impairment (GFR < 30 mL/min/1.73 m^2^) [27]. After 4 to 6 weeks, therapeutic drug monitoring is highly recommended [27].

In the present case, we decided to treat the patient with dalbavancin at 500 mg per week for 12 weeks after a 1000 mg loading dose. According to the Swedish national guidelines, three months of antimicrobial therapy is still recommended (www.infektion.net, accessed on 30 November 2022). Treatment with vancomycin, linezolid, and daptomycin had already been prescribed for some weeks, and these weeks could probably have been removed from the total treatment time. However, the course was protracted, and the local inflammation decreased only gradually. The tolerability and safety profile of dalbavancin were both good. No adverse reactions were reported by the patient, and laboratory tests and examination revealed no pathological alterations of the laboratory parameters, including haematology, liver tests, and renal function examination. Moreover, the increase in serum creatinine and decrease in GFR observed during treatment with vancomycin was normalized during the long-term treatment with dalbavancin.

Determination of the concentration of dalbavancin was performed only once, after the first loading dose, and since the trough level was adequate no additional concentrations were checked. It would unquestionably have been of great interest to follow the trough levels during this prolonged treatment. However, there was no laboratory facility providing this service in Scandinavia. Determination of serum bactericidal effect has been reported as an option for determining the dosing intervals during long-term treatment with dalbavancin [28].

*C. striatum* is regularly reported as multidrug-resistant [14,29], with a phenotype resistant to penicillins, carbapenems, aminoglycosides, fluoroquinolones, lincosamides, and macrolides, but susceptible to glycopeptides, tigecycline, daptomycin, and linezolid. This is in accordance with the antibiotic susceptibility pattern of the isolates from our case. PAP-AUC revealed no indications of presence of heteroresistance. These experiments did show the presence of suspected small colony variant (SCV) phenotypes, but these SCVs did not display any difference in antibiotic susceptibility to dalbavancin.

When analysing the genomes, the four isolates were almost identical, with two and five SNPs among the two normal and two small colony isolates, respectively, and only six SNPs separating any two isolates. Comparing these isolates to publicly available genomics data from RefSeq (Figure 2) showed that the four PJI isolates were distinct; however, due to the limited availability of genomic analyses of virulence factors, no inference of increased virulence potential of the Swedish isolates was possible. Multidrug-resistant *C. striatum* has been reported in the literature, but our isolates only carried two known resistance genes conferring resistance to tetracycline and macrolides.

Production of biofilm by bacteria is regarded as an important virulence factor, especially in the establishment of prosthetic device infections. Staphylococci are well recognized as having the ability to form biofilm, but *Corynebacterium* species, including *Corynebacterium striatum*, have also been reported to be biofilm producers [30]. What may be interesting is the fact that dalbavancin in in vitro experimental models has shown activity against biofilm-embedded Gram-positive bacteria [31,32] and has been proposed as effective therapy for these infections [21]. The mechanism is unclear, but seems to include both inhibition of biofilm formation and activity against bacteria within biofilm.

## 5. Conclusions

Fulminant early post-interventional PJI due to *C. striatum* can successfully and safely be treated by DAIR and long-term (12 weeks) dalbavancin therapy.

## Figures and Tables

**Figure 1 microorganisms-11-00550-f001:**
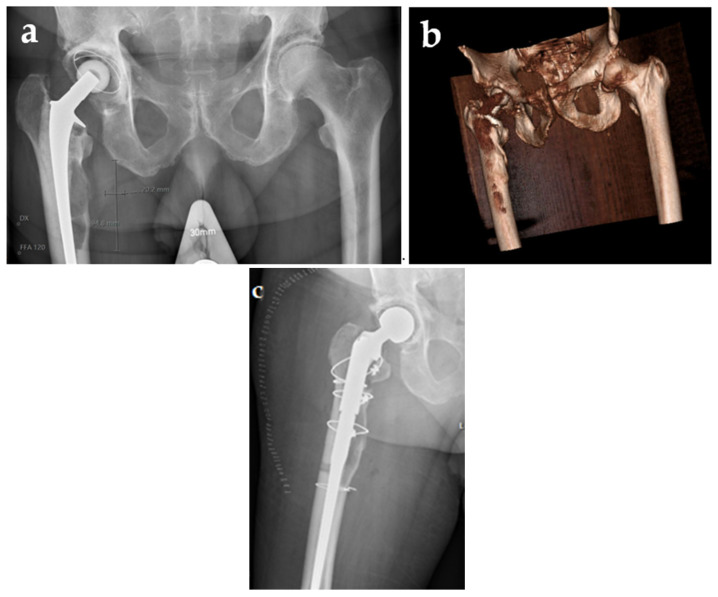
Preoperative X-ray (**a**), CT scan (**b**), and postoperative X-ray of one-stage revision arthroplasty (**c**).

**Figure 2 microorganisms-11-00550-f002:**
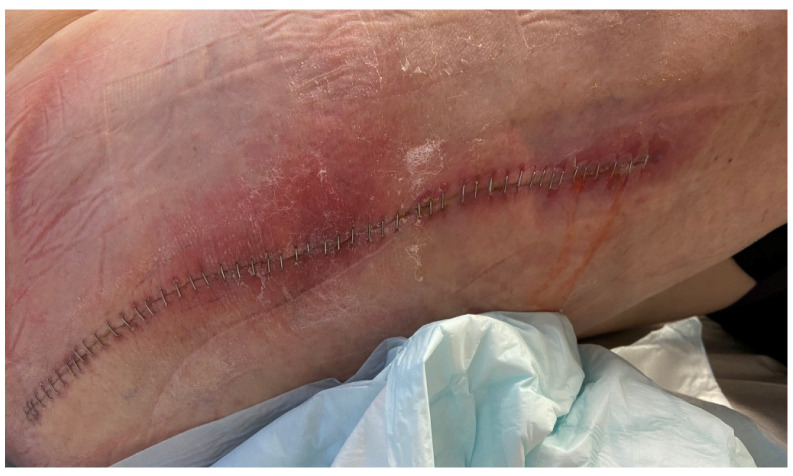
Surgical wound three weeks after one-stage revision surgery.

**Figure 3 microorganisms-11-00550-f003:**
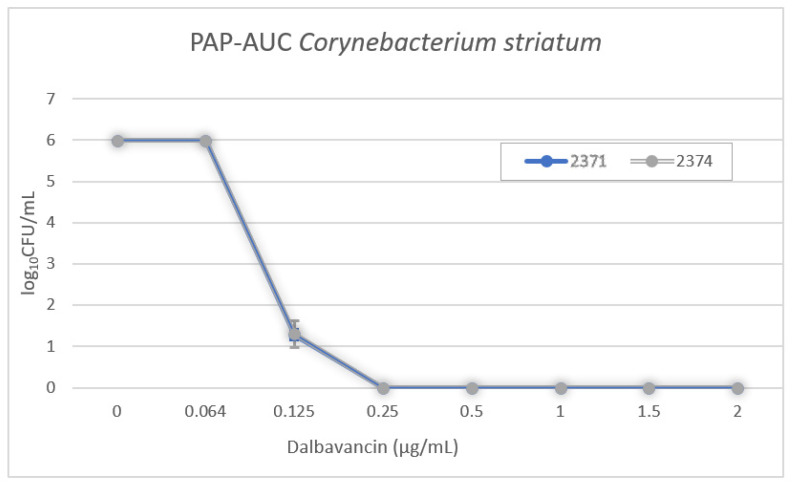
Population analysis profile–area under the curve (PAP-AUC). The two graphs represent the change of growth of total colonies (log_10_CFU/mL) at different antibiotic concentrations (μg/mL) of dalbavancin. Error bars indicate standard deviation.

**Figure 4 microorganisms-11-00550-f004:**
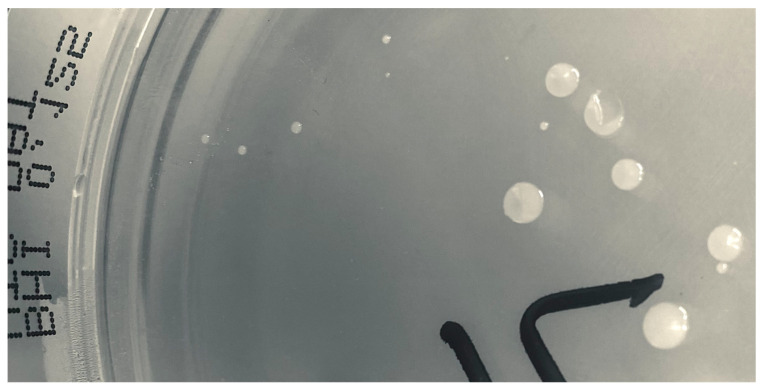
Growth of *Corynebacterium striatum* on BHI agar containing 0.125 μg/mL of dalbavancin, showing both normal-sized and tiny colonies.

**Figure 5 microorganisms-11-00550-f005:**
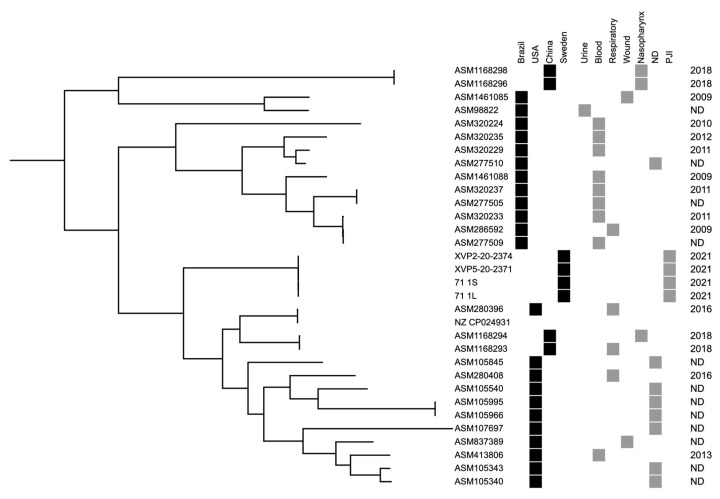
Midpoint rooted maximum likelihood phylogeny of all available *Corynebacterium striatum* genomes of human origin in the NCBIs RefSeq database and two isolates (2371 and 2374) obtained from a patient with a prosthetic joint infection described in the present paper, and the comparison between the normal-sized (2371S) and tiny colonies (2371L). Annotations show the year of isolation, source of the isolates, and country of origin. NA: Not available. Scalebar indicate substitutions per site.

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
