# Peer review of "Corynebacterium striatum Prosthetic Joint Infection Successfully Treated with Long-Term Dalbavancin"

_microorganisms, 2023, doi:10.3390/microorganisms11030550_

Round 1

Reviewer 1 Report

The authors aimed to present a case with a fulminant early post-interventional infection due to C. striatum following one-stage revision surgery that was successfully treated with debridement, antibiotics, and implant retention (DAIR) followed by long-term dalbavancin therapy.

The study covers some issues that have been overlooked in other similar topics. The structure of the manuscript appears adequate and well divided in the sections. Moreover, the study is easy to follow, but some issues should be improved. Some of the comments that would improve the overall quality of the study are:

I-) Authors must pay attention to the technical terms acronyms they used in the text

II-) Conclusion Section: This paragraph required a general revision to eliminate redundant sentences and to add some "take-home message".

Author Response

Thank you for the comments.

1) All acronyms such as PJI, DAIR, MIC, PAP-AUC and MALDI TOF MS has been spelled out first time used. However, company namnes have not been spelled out. 

2) The conclusion has been revised and is highlighted in the revised Ms

Reviewer 2 Report

Estimated authors,

Your manuscript present a very interesting case of a common medical condition (PJI). Several features make this case specially interesting. First, an exhaustive, thorough and brilliant microbiological analysis of the microorganism involved. Second, the choice of Dalbavancin as a treatment option for this patient. 

Nevertheless, there are some issues which, from my point of view, should be improved or require further discussion. First: In the discussion section, I would like to know why you chose this dosing scheme. As you know, there is not a standard scheme accepted for DAL for treating PJI. Besides, so as to 2021, very few cases of PJI managed with DAIR + DAL, had been reported, and there is a black-hole sized gap pf knowledge in this concrete aspect of PJI management. This has been extensively reviewed so as to 2021 (Buzon-Martin L et al , Antibiotics (Basel) 2021), and I think you should discuss thoroughly why you chose that dosification scheme according to the previous (almost inexistant) lack of evidence.  In order to improve this aspect, I consider that the original papers of Dunne et.al regarding DAL PK/PD are essential to defend the dosing scheme you chose (Dunne MW, AAC 2015)

The determination of DAL plasma levels is a very interesting issue. So as to now, we have very few data about it. There are some ongoing studies trying to shed some light on this. You mention you made one determination, but do not provide values, and do not mention it in matherial and methods. You should add it to M&M section, and discuss it in the Discussion section as well, and I suggest you to use some specific references when necessary (specifically Dunne MW AAC 2015 and others at your discretion)

Another minor considerations:

- Minor english language spell check required.

Looking forward to seing the new version

Author Response

Thank you for the comments.

1) A short paragraph has been added in the discussion section and the suggested references have been added. Please see the added wording (in red) in the revised manuscript. 

When the patient was treated some years ago the loading dose was reduced to 1,000 mg due to previous renal impairment, followed by 500-mg weekly doses for in total 12 weeks following the DAIR procedure according to the national guidelines, as mentioned in the manuscript. Today I have started with 2 loading doses and then 1,000 mg biweekly. 

2) Only one determination of the dalbavancin concentration was perform and that was after the first dose. The sample was sent on ice by air  to Nantes since no such facility was (are) present in Sweden. Of course, TDM had been valuable in this case. The result of the analysis and the laboratory in Nantes is mentioned in the manuscript. 

- Minor spelling errors have been corrected and are highlighted in red.